# Polysaccharide-Peptide from *Trametes versicolor*: The Potential Medicine for Colorectal Cancer Treatment

**DOI:** 10.3390/biomedicines10112841

**Published:** 2022-11-07

**Authors:** Zhicheng He, Jian Lin, Yingying He, Shubai Liu

**Affiliations:** 1State Key Laboratory of Phytochemistry and Plant Resources in West China, Kunming Institute of Botany, Chinese Academy of Sciences, Kunming 650201, China; 2University of Chinese Academy of Sciences, Beijing 100049, China; 3School of Chemical Science & Technology, Yunnan University, Kunming 650091, China

**Keywords:** *Trametes versicolor*, polysaccharide peptide, colorectal cancer, traditional medicine

## Abstract

The incidence and mortality of colorectal cancer have shown an upward trend in the past decade. Therefore, the prevention, diagnosis, and treatment of colorectal cancer still need our continuous attention. Finding compounds with strong anticancer activity and low toxicity is a good strategy for colorectal cancer (CRC) therapy. *Trametes versicolor* is a traditional Chinese medicinal mushroom with a long history of being used to regulate immunity and prevent cancer. Its extractions were demonstrated with strong cell growth inhibitory activity on human colorectal tumor cells, while the anticancer activity of them is not acted through a direct cytotoxic effect. However, the intricacy and high molecular weight make mechanistic research difficult, which restricts their further application as a medication in clinical cancer treatment. Recent research has discovered a small molecule polysaccharide peptide derived from *Trametes versicolor* that has a distinct structure after decades of *Trametes versicolor* investigation. Uncertain molecular weight and a complex composition are problems that have been solved through studies on its structure, and it was demonstrated to have strong anti-proliferation activity on colorectal cancer in vitro and in vivo via interaction with EGFR signaling pathway. It opens up new horizons for research in this field, and these low molecular weight polysaccharide peptides provide a new insight of regulation of colorectal cancer proliferation and have great potential as drugs in the treatment of colorectal cancer.

## 1. Introduction

Cancer is the second leading cause of mortality in the world and with 9.9 million deaths in 2020. Colorectal cancer (CRC) is the second leading cause of malignant tumor-related deaths and the third most diagnosed cancer in the world [1]. According to data from Globocan in 2020 [2], there are 1.93 million new cases of CRC and 935,175 people that die from CRC. Compared with 2012 [3], the case numbers were increased about 330,000 and 240,000 people, respectively. Therefore, the prevention, diagnosis, and treatment of colorectal cancer still need to be continuously attended. At present, the major clinical treatments for colorectal cancer are surgery and chemotherapy, while natural products and their derivatives play important roles in chemotherapy. Natural products are compounds that are extracted from plants, animals, and microorganisms. The diverse pharmacological activities of natural products are based on the complexity and diversity of their structures and also provide numerous lead compounds for the discovery of new drugs. Natural product researchers have been searching for compounds that are effective at killing tumor cells but more friendly to normal cells in last few decades due to most types of natural products inhibiting the proliferation of cancer cells by their highly cytotoxic nature, and their side effects affect patients’ quality of life.

Traditional Chinese medicine uses *Trametes versicolor* (formerly *Coriolus versicolor*) for its longevity-enhancing and health-promoting properties. Polysaccharide Krestin (PSK) or polysaccharide peptide (PSP) are two natural products extracted from *Trametes versicolor*, and their main components are a highly heterogeneous mixture of β-glucan macromolecules that possess a molecular weight of approximately 100 kDa and contain various moieties, including peptides, bound to β-glucan backbones [4,5,6]. They have been used as adjuvant therapy for cancer in Japan and China [7] and are commonly considered as nontoxic in addition to having no adverse effects [8]. The complexity and high molecular weight of PSK and PSP, however, make it difficult to study their mechanism of action [9], and for a long time, they have only been used as adjuvant to supplement chemotherapy and radiation therapy rather than as an anti-cancer drug for clinical treatment. In 1992, Yang et al. [10] isolated a small polypeptide with a molecular weight of around 10 KDa from the crude extraction of *Coriolus versicolor* (Cov-1) polysaccharide peptide, and its anti-cancer activities were significantly higher than PSP and PSK. A polysaccharide–peptide complex with a molecular mass of approximately 15.5 KDa was discovered by Wang et al. in 1996 [11] isolated, and it possessed the activities of inhibiting the growth of mice-implanted sarcoma 180 cells. The results of this study advanced the research on *Trametes versicolor* glycopeptide. However, the structure of these small molecular weight glycopeptides was still unclear at the time due to the technical limitations, and no more studies on pharmacological mechanisms were conducted. Musarin, a novel 12 kDa polysaccharide peptide isolated from *Trametes versicolor* powder, was recently found and described by He et al. [12]. Its protein sequence was verified, and its 3D structure was predicted in this study. Studies on the associated mechanism have shown that musarin has potent anti-colon cancer activity without being harmful to healthy colon cells in both in vivo and in vitro tests. Our understanding of traditional Chinese medicine has grown as a result of this research, and these findings imply that *Trametes versicolor* polysaccharide peptides provide a new insight of regulation of colorectal cancer proliferation and have great potential as drugs in the treatment of colorectal cancer.

This paper mainly summarized the history of research progress on *Trametes versicolor* polysaccharide peptides’ anti-colorectal cancer activity and related developments. We also propose a few directions for further research into these glycopeptides and their application value in the treatment for colorectal cancer and other malignancies.

## 2. PSK and PSP: Mainly Polysaccharide Peptides in *Trametes versicolor*

Due to its wide range of adhibition as a food supplement in western countries, *Trametes versicolor* has experienced growth in popularity. The primary products are polysaccharide Krestin (PSK) and polysaccharide peptide (PSP), which were extracted from different strains of *Trametes versicolor,* including “CM-101” (PSP in China) and “Cov-1” (PSP Krestin or PSK in Japan) [8]. Their main components are a highly heterogeneous mixture of β-glucan macromolecules with a molecular weight of about 100 kDa that contain various moieties, including peptides, bound to their backbones [4,5,6]; the contents of other sugar components such as fucose, galactose, mannose, and xylose were different [7,13]. Their differences are mainly in peptide content and glycan composition; PSP contains about 10–30 percent peptides, while PSK contains up to 90 percent peptide [14]. Due to their complex compositions, its precise structure cannot be clarified. All that can be determined is the main molecular structure of the polysaccharide component [13]. Additionally, they have received medical approval primarily for use as adjuvants in the treatment of cancer in China and Japan. The immunomodulatory effects of polysaccharide peptides may be responsible for this. PSK was approved for marketing in Japan in 1977 after a long period of clinical trials. In 1987, PSK had annual sales of $357 million in Japan [4]. The PSP was applicated around the 1990s, a decade after the PSK. Studies have shown that PSP has a variety of physiological functions such as enhancing immunity, anti-tumor, protecting the liver, anti-oxidation, and lowering blood lipids. PSP has been clinically used to treat cancer, hepatitis, hyperlipidemia, chronic bronchitis, and other diseases. Moreover, PSP has low cytotoxicity and no serious adverse reactions [8], but also can enhance the therapeutic effect of chemoradiotherapy drugs [15], alleviate the toxicity and side effects caused by chemoradiotherapy in cancer patients, and improve the quality of life of patients [16].

## 3. Anti-Tumor Activity of *Trametes versicolor* Extracts

Previous studies indicated that both PSK and PSP have a variety of anti-cancer activities. They are effective whether administered by orally, intravenously, or intraperitoneally, and are frequently regarded as nontoxic as well as free of adverse effects [8]. Numerous studies have showed the potent and widespread antitumor activities of PSP and PSK. PSK exhibited anti-leukemic [17], anti-hepatoma [18], anti-ovarian cancer [19], anti-breast cancer [20], and anti-colorectal cancer cell [21,22,23] activities, according to in vitro experiments. Additionally, PSP has anti-glioma [24], anti-HL-60 cells [25], anti-prostate cancer [26], anti-breast cancer [27], and anti-HepG2 cells [28] activities. Furthermore, PSK and PSP also exhibit anti-tumor potential in vivo, as seen in cases of prostate cancer [22,26], hepatoma [29], and mammary gland tumors [30]. 

Human leukemia HL60 cell is the most commonly utilized to examine the anti-tumor effect of PSP because they can inhibit the proliferation of tumor cells by disrupting the malignant cell cycle [25,31] and inducing apoptosis [14]. The anti-apoptotic proteins Bcl-2 and Survivin were shown to be decreased during these effects, while Bax and cytochrome C are increased. A number of phosphatase and kinase genes are also activated, along with caspase-3, -8, and -9 [32]. PSP significantly enhanced radiation-induced in vitro damage to C6 rat glioma cells [24]. According to results of Luk et al. [26], PSP treatment of prostate cancer PC-3 cell lines led to a time- and dose-dependent downregulation of CSC (cancer stem cell) markers (CD133 and CD44). In addition to inhibiting PC-3 cells to form prostatic spheres, PSP treatment significantly reduced the tumorigenicity of the cells in vivo. Furthermore, it was demonstrated that oral administration of PSP significantly reduced prostate tumor formation in TgMAP mice and inhibited prostate cancer tumor stem cell proliferation. PSP inhibit the growth and metastasis of various tumors in animal models in addition to preventing tumor formation brought on by different chemical carcinogens [15]. PSP inhibited cancer cell migration and significantly reduced the production of the matrix metalloproteinase MMP-9 in a time- and dose-dependent manner, according to in vitro cell migration assays of breast cancer cells from 4T1 mice with PSP. When given PSP therapy in vivo, mice injected with 4T1 cells showed inhibited proliferation in the lung, preventing the development of liver metastases [33]. Cancer cells treated with PSP had longer DNA synthesis time (Ts), while Doxo and VP-16 indicated stronger pro-apoptotic effect [27]. PSP enhanced the plasma half-life of anti-cancer drugs and decreased the clearance rate of cyclophosphamide, while increasing the cytotoxic effect of cyclophosphamide on the HepG2 cancer cell line [28].

These findings indicated that polysaccharide peptides in *Trametes versicolor* had potent antitumor activity, provide a wealth of references for PSK and PSP research on anticancer studies, and highlight their potential for thorough research and cancer treatment.

## 4. Anti-Colorectal Cancer Activity of *Trametes versicolor* Extraction

PSK was found to have an inhibitory effect on colon cancer cell lines (HT29 and SW480) by Hirahara et al. [21,34], but additional research was not done. According to Roca’s results [23], LoVo and HT-29 human colon cancer cells were prevented from proliferation, migration, and invasion by treatment of polysaccharide-rich extracts from *Trametes versicolor*. They suggested that the antitumor activity may be caused by elevating the expression of E-cadherin protein and suppressing the activity of MMP-2. Additionally, Knezevic [35] discovered that the extraction (basidiocarps and mycelium) from *Trametes versicolor* exhibited the activity against the proliferation of human colon carcinoma (LS174). Aside from these studies, few studies have addressed the anti-colorectal cancer activity of PSK and PSP. This may be attributed to the large molecular weight and complexity of PSP, which makes the future of its research prospects not very promising.

## 5. Polysaccharide Peptides with Smaller Molecular Weight from *Trametes versicolor*

Researchers have also tried to determine whether there are smaller polypeptides that exert anti-cancer activity potential given the large molecular weight of PSP. In 1992, Yang et al. [10] isolated a small polypeptide known as SPCV with about 10 KDa molecular weight from the crude extraction of polysaccharide peptide of *Coriolus versicolor* (Cov-1). Additionally, in vitro experiments indicated that the proliferation of leukemia cells and SCG-7901 were significantly more inhibited in SPCV treated group than that in PSP and PSK groups. In nude mice inoculated with tumor cells, pretreatment of SPCV for two weeks significantly decreased the incidence of tumor mass. From the mycelial culture of *Tricholoma mongolicum*, Wang et al. [11] also isolated a polysaccharide-peptide complex with a molecular mass of 15.5 kDa in 1996. This complex possessed the activities of activating macrophages, stimulating macrophage antigen-presenting, which in turn enhanced proliferation of T-cells, and inhibiting the growth of sarcoma 180 cells that had been implanted in mice. 

These findings imply that smaller molecular weight polypeptides in PSP might be the active substances. Additionally, they have a greater chance of being used as a drug in the world than PSK and PSP, which have heavier molecular weights. Physical and chemical properties revealed them to be mixtures with smaller molecular weight peptides, but their sequences and structures were not clearly clarified, possibly as a result of the technical constraints of the period. This makes it challenging to perform further research on the pharmacological mechanisms that exert their anticancer activity. Additionally, it restricts its ability to be used clinically in the treatment of colorectal cancer.

## 6. Musarin: A Novel Polysaccharide Peptide from *Trametes versicolor*

PSK and PSP’s complicated composition and high molecular weight make mechanistic research difficult, which hinders their adoption into Western medicine’s pharmacopeia [9]. Therefore, determining PSK and PSP’s active components is very important for further study because of this. Recent research has shed new light on the polysaccharide peptide from *Trametes versicolor*. In this study, a novel 12 kDa protein named musarin was discovered and characterized, which was isolated from *Trametes versicolor* powder [12]. Multiple colorectal cancer cell lines and functional assays were used to assess the anti-cancer bioactivity of musarin. The findings indicated that musarin inhibits the proliferation of multiple colon cancer cell lines, while it did not influence the growth of normal colon cells. Additionally, it did not cause cancer cells to undergo apoptosis and necrosis. It is suggested that musarin prevents proliferation of colorectal cancer cells without having lethal effects. Furthermore, colorectal cancer stem cells and a NOD/SCID murine xenograft model were used to assess its anti-cancer effectiveness. The size and weight of tumor xenografts in NOD/SCID mice were significantly suppressed in vivo after receiving musarin orally for 14 days. Additionally, oral administration of musarin had similar impact to gefitinib in clearing tumors. Moreover, none of the musarin-treated mice had the skin rash and hair loss side effects that some gefitinib-treated mice did. 

Musarin’s mechanism of action has also been studied due to its clear structure and strong anti-tumor activity. Firstly, musarin indicated dose-dependent EGFR tyrosine kinase inhibitor activity, with an IC_50_ value of 1.39 μM. Musarin is worked to function as tyrosine kinase inhibitor to suppress the growth of colorectal cancer stem-like cells and does not have the same adverse effects as gefitinib. Musarin treatment downregulated the EGFR-Ras signaling pathway (Figure 1A) in colorectal cancer stem-like cells, including EGFR, p-Akt (Ser473), and cyclin D1, as well as p-EGFR-Y1173 and p-EGFR-Y1045. As a result, the EGFR signaling pathway is quenched by musarin-modulated phosphorylation of specific EGFR sites, which inhibits overall EGFR activity limiting proliferation of the CSC-like CD24^+^CD44^+^ HT29 subpopulation. Musarin is expected to inhibit proliferation and epithelial to mesenchymal transition of colon cancer cells via altering the EGFR phosphorylation selectivity and differential expression of other related signaling molecules. This is the first time the molecular mechanism of *Trametes versicolor*-derived polysaccharide peptides’ inhibitory effect on colorectal cancer has been properly reported. This study not only provides the groundwork for further research into the specific structure of the *Trametes versicolor* polysaccharide peptide, but it also demonstrates that it has significant therapeutic promise in the treatment of colorectal cancer.

## 7. Immunoregulatory Effect of Polysaccharide Peptide

Numerous studies have shown that PSP demonstrates an immunoregulatory effect and regulates the functions of multiple immune cell types. This may be an indispensable part of its anti-tumor activity. Human PBMCs treated with PSP showed a time-dependent increase in proliferative responses; they were found to secrete more TNF-α than group without PSP treatment [36]. When healthy mice are orally administered PSP, their peritoneal macrophages respond to in vitro stimulation by producing of reactive nitrogen intermediates and superoxide anions [37]. Regardless of the presence or absence of phytohemagglutinin (PHA) [27], TNF-α increased more than 3.5-fold when PSP was incubated with PBMC derived from healthy individuals. When PSP was applied to PBMC from breast cancer patients, TNF-α secretion increased as well. This effect was not abolished by TLR4 blockade, indicating that PSP is not dependent on TLR4 activation [38]. The activation of IL-12, a T helper cell 1 (Th1)-associated cytokine that enhances the cytotoxic activity of NK and CD8+T cells and their TNF-α expression, further confirmed PSP’s capacity to induce cytokines linked with TNF-α. A threefold increase in IL-12 expression was seen in mouse splenic lymphocytes with CV extract for 48 or 72 hours compared with control [39]. PSP affects cytokine release by immune cells, as well as making them more sensitive to external stimuli, and acts synergistically with other factors [40]. Our most recent studies demonstrated that PSP pretreatment significantly decreased the expression of downstream molecules PD-L1 and EGFR signaling pathways (c-Jun and STAT3) in HCT116. Additionally, pretreatment with polysaccharide peptide enhanced the T-cells’ killing effect induced by co-culture PBMC on HCT116 cells [41].

The blood and spleen levels of NK cells, lymphocytes, and granulocytes were significantly increased in PSP-treated rats, indicating that PSP has a protective effect on the immune system [24]. A protective effect of PSP on external death signals [42] has been shown by 48-hour culture with PSP that decreased Fas receptor expression in unstimulated lymphocytes and synergistically acted with cyclosporine to produce similar effects on PHA-stimulated cells. Monocytes and macrophages can also be impacted by PSP. A 48-hour PSP treatment on healthy human PBMC resulted in an increase in CD14^+^CD16^-^MHCII^+^ monocytes [43]. However, some studies have shown that PSP boosts macrophages phagocytic activity both in culture and in vivo [44]. At the same time, humoral immunity was significantly impacted by the C. lucidum glycopeptide. PSP can stimulate the secretion of IgM and the activation of B cell population in mouse spleen cells, and the authors hypothesized that it may activate mouse B cells through MAPK and NF-κB signaling pathways [45]. PSP can boost NK cytotoxic function through IL-18 induced upregulation of FasL expression [14]. TRAM/TRIF/TRAF6 signaling pathway of TLR may be one of the key signaling pathways connected to PSP immune regulation [46], specifically gene expression and cytokine secretion related to TLR signaling pathway in PBMC regulated by PSP. However, multiple genes, kinase phosphorylation, and protein levels in the TLR4 pathway of PBMC in patients were significantly up regulated after PSP treatment of breast cancer patients [47]. Hsu et al. [48] discovered through clinical trials that PSK could prolong the survival of patients with stage IIIA/IIIB gastric cancer, especially in patients with PD-L1(-) subgroup. This strengthens the possibility that PSP has equivalent capabilities. 

PSP has a strong immunomodulatory effect that not only enhances the immune ability of the body but also directly exerts anti-tumor activity. Therefore, the potency of this combination means PSP has great application potential in cancer treatment, as evidenced by the combination of the immune-stimulating action and its capacity to repress proliferation of cancer cells. 

## 8. Discussion

Finding an effective treatment for colorectal cancer (CRC) is particularly important and urgent since the number of CRC cases and fatalities is rising every year, and CRC is the second leading cause of malignant tumor-related deaths and third most diagnosed cancer worldwide. At the moment, the primary clinical treatments for colorectal cancer are chemotherapy and radiotherapy. Especially, patients who have undergone surgery as well as those who unable to have surgery due to metastases at the time of diagnosis need chemotherapy. In the therapeutic treatment of colorectal cancer, the chemotherapy drugs 5-fluorouracil [49,50], oxaliplatin, and irinotecan [51,52] could be administered alone or in combination [53]. Since tumor cells might grow resistant to or insensitive to a single chemotherapy agent, the idea of medication combination has been put up. These chemotherapy drugs play an important role in the treatment of colorectal cancer, but because of their cytotoxic activity, they also seriously harm patients’ normal cells, resulting in many side effects, such as alopecia, general weakness, effects on family/partner, loss of taste, nausea, fatigue, difficulty sleeping, effects on work/home duties, as well as the need to put life on hold [54,55]. Patients’ quality of life is extremely poor since they must endure psychological stress and great physical discomfort while undergoing treatment. 

In addition to being a source of drugs, medicinal mushrooms are now being used as adjuvants with conventional chemo- or radiation-therapy to either enhance their potency or reduce their bad effects [7,56]. *Trametes versicolor*-derived PSK and PSP are both efficacious when administered orally, intravenously, or intraperitoneally, and they are commonly considered to be nontoxic and free of side effects [8]. In China, PSK, whose market value was USD 357 million, became a clinically used drug in 1987 [57] after receiving clinical use approval in Japan in 1977. Additionally, earlier studies have indicated that PSK and PSP have potent anti-cancer activities in a variety of cancer situations, including leukemic, breast, cervix, stomach, liver, lung, prostate, ovarian, and colorectal cancer [6,8,15,20,22,23]. However, PSP’s huge molecular weight and complexity makes it difficult to understand how it works and restricts how broadly it may be used as a clinical treatment. 

In 1992, Yang et al. [10] isolated a small polypeptide known as SPCV, with about 10 KDa molecular weight, from the crude extraction of polysaccharide peptide of *Coriolus versicolor* (Cov-1). A polysaccharide–peptide complex with a molecular mass of 15.5 kDa was obtained in 1996 by Wang et al. [11] from the *Tricholoma mongolicum* mycelial culture. In vitro and in vivo tests indicated that these small weight polypeptides had significantly stronger inhibitory activity than PSP and PSK. These results imply that the active substances in PSP might be smaller molecular weight polypeptides. Additionally, they have a greater chance of being used as a drug around the world than PSK and PSP. These results indicate the importance of carefully and precisely examining the antitumor activity and mechanism of these compounds.

The latest research results explained the mechanism of its anti-colon cancer effect in addition to resolving the challenge of structure and sequence analysis. He and his colleagues [12] identified a novel 12 kDa polypeptide named musarin from *Trametes versicolor* powder. The polypeptide was discovered as having a cytoplasmic and transmembrane domain, and sequencing results revealed that it had 112 amino acids. Additionally, ORION (optimized protein fold recognition) was used to predict the 3D structure of musarin. The results of these structural studies serve as a basis for the investigation of anti-colon cancer activities and behind regulatory mechanisms. The results showed that musarin inhibited multiple colorectal cancer cells proliferation without causing any damage to healthy colon cells and did not induce apoptosis or necrosis in cancer cells. It highlights the benefits of minimal cytotoxicity that conventional traditional drugs do not have. Moreover, oral administration of musarin for 14 days reduced tumor xenograft in NOD/SCID mice in vivo, and its effectiveness was comparable to that of gefitinib, the representative EGFR inhibitor. The results of the mechanism study imply that musarin inhibits proliferation of colonic carcinoma cells and the epithelial to mesenchymal transition via altering the selectivity of EGFR phosphorylation and by differentially expressing other related signaling molecules that are connected to this process. The findings of this study show that polysaccharide peptide from *Trametes versicolor has* great potential for treating colorectal cancer and serve as a foundation for further research into its precise structure.

We speculate that different short polysaccharide peptides are present in PSP and PSK. Furthermore, numerous studies have shown its capacity to inhibit a variety of cancer cells. The research methodology advance will offer guidance for examining PSP and PSK composition and structure and aid in the identification of additional active components or compounds that may be useful in treating colorectal cancer or other cancers. Because it acts on the EGFR signaling pathway, musarin may potentially be effective against other types of cancers, including lung and breast cancer. Medicines that target EGFR heavily rely on EGFRs. As a result, it could be useful for the study into various cancer types. 

Additionally, immunosuppression is the key feature of the immune microenvironment in colorectal cancer as common knowledge. There are numerous justifications for immune evasion, including immunoregulatory cells’ recruitment, upregulating inhibitory molecules (including macrophages, T regulatory cells, myeloid derived suppressor cells and so on), and downregulating antigen presentation [58]. Immunotherapies have been recognized as one of the most promising therapies in cancer treatment. PSP can aid in the body’s ability to reactivate its immune system and enhance immunological function [15]. Therefore, musarin might have immunostimulatory effects in addition to its ability to prevent tumor growth. According to the results of Lu et al. [59], PSK is a TLR2 agonist that inhibits tumor growth via stimulating CD8 T cells and NK cells. PSK is still a mixture, and its molecular mechanism is not clear. According to this study, musarin may also exert immunostimulatory effects in addition to acting on EGFR by other antagonistic receptors such as TLR2. The destination of tumor immunotherapy is to activate or enhance the body’s immune system so that it can recognize and kill tumor cells. Tumor cells can express PD-L1 on their surface, which enables them to avoid recognition and killing by immune cells and even induce apoptosis of immune cells [60]. PD-L1 is regulated by multiple signaling pathways and transcription factors in tumor cells, including p53, PETN, STAT3, and EGFR. Therefore, inhibitors targeting these signaling pathways can restore the immunosuppressed tumor microenvironment in patients, allowing immune cells to re-recognize and kill tumor cells while minimizing immune-related negative effects. Additionally, our study’s results indicated that PSP can suppress the expression of EGFR and PD-L1 [41]. By examining the important signaling molecules linked to EGFR and PD-L1, results showed that PSP treatment significantly reduced the expression levels of p-EGFR(Tyr1068), c-Jun, STAT3, and p-STAT3(Tyr705) in colorectal cancer cells. It is suggested that PSP deregulates PD-L1 expression by decreasing the expression of EGFR. Additionally, PSP can enhance the T-cells killing effect on colorectal cancer cells that induced by PBMC co-culture. It is proposed that PSP control additional signaling pathways or key regulators (Figure 1B). It is predictable that musarin will enhance immunity through these pathways as well and be worth to explore the potential molecular mechanism in future study.

The majority tumor treatments in the past prioritized “tumor-free survival”, and they were all directed towards shrinking the tumor’s size and attempting to eradicate all the tumor cells, which was counterproductive. Even while some curative benefits were achieved, they were all at the expense of the body and immune system’s health, and tumor metastasis and recurrence are still a possibility. Furthermore, after receiving “tumor-free” treatment, many patients’ quality life was reduced, even to the point where it resulted in “tumor does not disappear, people die first” or “tumor survival death” outcomes. Additionally, we think that polysaccharide peptide from *Trametes versicolor* may prolong the lifetime of cancer patients who have tumors by stopping the growth, invasion, or metastasis of the tumor cells. This is essential to reduce complications, enhancing patient quality of life, and extending survival time. These polysaccharide peptides may someday find useful applications (Figure 2).

In conclusion, *Trametes versicolor* polysaccharide peptides have good anti-colorectal cancer activity, show no toxicity to normal cells, and enhance patients’ ability to fight cancer through their immunostimulatory effects. The innovative small polysaccharide peptide musarin exhibits great potential as a promising drug for colorectal cancer treatment. In the future, these peptides should be studied thoroughly and in-depth for their potential in cancer treatment.

## Figures and Tables

**Figure 1 biomedicines-10-02841-f001:**
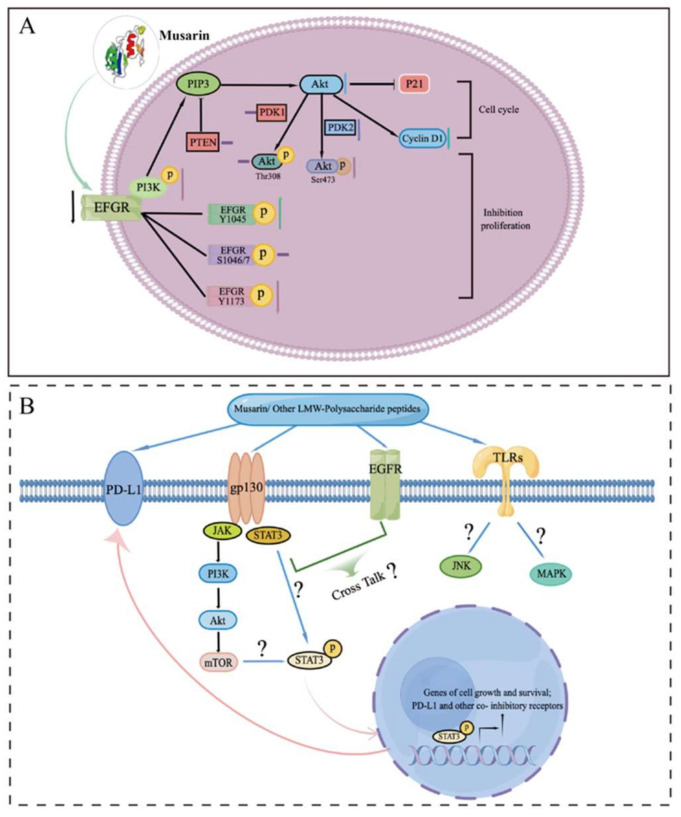
(**A**) Proposed molecular mechanism of *Trametes versicolor*-derived polysaccharide peptides’ musarin inhibitory effect on colorectal cancer. (**B**) Potential anti-proliferation molecular mechanism of musarin and other identified low molecular weight polysaccharide peptides (LMW-polysaccharide peptides) on CRC cells. (“?” Represents a mechanism that is currently unclear and need to be studied in the future).

**Figure 2 biomedicines-10-02841-f002:**
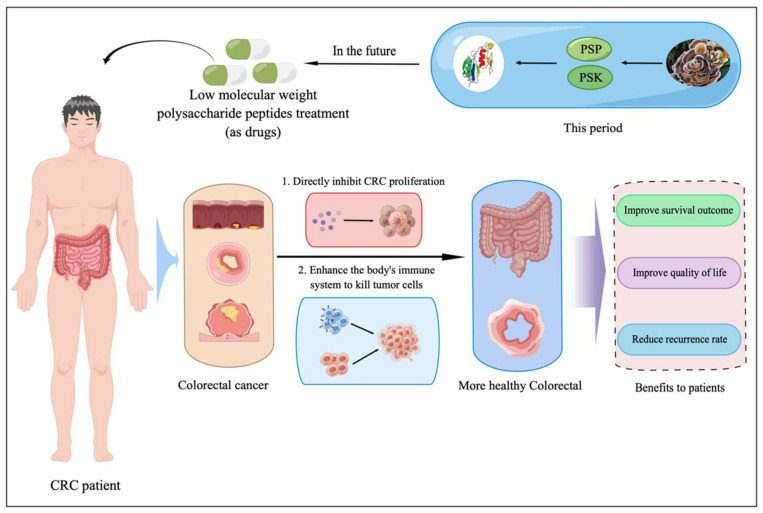
Prospects of musarin (the 3D structure was derived from our previous study [12]) and LMW-polysaccharide peptides applicated for CRC patients’ treatment.

## Data Availability

Not applicable.

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
