# Peer review of "Polysaccharide-Peptide from *Trametes versicolor*: The Potential Medicine for Colorectal Cancer Treatment"

_biomedicines, 2022, doi:10.3390/biomedicines10112841_

Round 1
Reviewer 1 Report
Reviewer: Recommendation: The Article review titled as “Polysaccharide-peptide from Trametes versicolor: the potential medicine for colorectal cancer treatment.” may be publishable, but it should be reviewed after minor revisions Comments: The review provides in depth information, the article is well written; the references are appropriate. Therefore, its publication in “Biomedicines” is recommended after minor revision. Additional comments: Page No, 2, Line 65, Keep the KDa format consistent throughout the article. Page No. 2, Line 66, “He et al15 .” remove the space Page No. 4, Line 192, “IC50” 50 need to be subscript in entire article. Page No. 5, Figure 1, Cleanup he figures 1A and B. Text are tiny and not readable clearly. Resolution is poor. Page No. 9, Figure 2, Cite the references from where the structure was taken from. In reference section please keep the journal writing style (as per Biomedicines) CONSISTENT for all the listed references. Text size throughout the review is not consistent, please correct accordingly
Author Response
Dear reviewer,
Our point-by-point response to the comments submit as PDF document. Please see the attachment.

Reviewer 2 Report
1- Abstract lacks the conclusion or recommendation of the review.
2- line 46. uses not italic.
3- line 82. including not italic.
4- line 100. extract(s) not extraction.
5- talking about PSK & PSP should be accompanied with some details about the structure/composition. illustrative figure will be preferably added.
6- narration of antitumor activity lacks mentioning the nature of solvents used in extraction, if they polar or non.
7- I see that this work should include some in vivo studies concerning the cytotoxicity, cytotoxicity on the in vitro scale is not sufficient. also, some immunological studies should be merged to elaborate clearly the action mechanism.
Author Response

(The authors gave the same response as above.)

Round 2
Reviewer 2 Report
The review article still needs more careful revision to valorize/maximize the benefit to the readers.